# Novel Identification of Ankyrin-R in Cardiac Fibroblasts and a Potential Role in Heart Failure

**DOI:** 10.3390/ijms25158403

**Published:** 2024-08-01

**Authors:** Aaron D. Argall, Holly C. Sucharski-Argall, Luke G. Comisford, Sallie J. Jurs, Jack T. Seminetta, Michael J. Wallace, Casey A. Crawford, Sarah S. Takenaka, Mei Han, Mona El Refaey, Thomas J. Hund, Peter J. Mohler, Sara N. Koenig

**Affiliations:** 1Frick Center for Heart Failure and Arrhythmia Research, Dorothy M. Davis Heart and Lung Research Institute, Wexner Medical Center, Ohio State University, Columbus, OH 43210, USA; aaron.argall2@nationwidechildrens.org (A.D.A.); holly.sucharski@osumc.edu (H.C.S.-A.);; 2Division of Cardiovascular Medicine, Department of Internal Medicine, Ohio State University, Columbus, OH 43210, USA; 3Department of Physiology and Cell Biology, College of Medicine, Ohio State University, Columbus, OH 43210, USA; 4Division of Cardiac Surgery, Department of Surgery, Ohio State University, Columbus, OH 43210, USA; 5Department of Biomedical Engineering, College of Engineering, Ohio State University, Columbus, OH 43210, USA

**Keywords:** Ankyrin-R, heart failure, fibroblast, fibrosis, cardiomyocyte

## Abstract

Altered ankyrin-R (AnkR; encoded by *ANK1*) expression is associated with diastolic function, left ventricular remodeling, and heart failure with preserved ejection fraction (HFpEF). First identified in erythrocytes, the role of AnkR in other tissues, particularly the heart, is less studied. Here, we identified the expression of both canonical and small isoforms of AnkR in the mouse myocardium. We demonstrate that cardiac myocytes primarily express small AnkR (sAnkR), whereas cardiac fibroblasts predominantly express canonical AnkR. As canonical AnkR expression in cardiac fibroblasts is unstudied, we focused on expression and localization in these cells. AnkR is expressed in both the perinuclear and cytoplasmic regions of fibroblasts with considerable overlap with the trans-Golgi network protein 38, TGN38, suggesting a potential role in trafficking. To study the role of AnkR in fibroblasts, we generated mice lacking AnkR in activated fibroblasts (*Ank1*-ifKO mice). Notably, *Ank1*-ifKO mice fibroblasts displayed reduced collagen compaction, supportive of a novel role of AnkR in normal fibroblast function. At the whole animal level, in response to a heart failure model, *Ank1*-ifKO mice displayed an increase in fibrosis and T-wave inversion compared with littermate controls, while preserving cardiac ejection fraction. Collagen type I fibers were decreased in the *Ank1*-ifKO mice, suggesting a novel function of AnkR in the maturation of collagen fibers. In summary, our findings illustrate the novel expression of AnkR in cardiac fibroblasts and a potential role in cardiac function in response to stress.

## 1. Introduction

Ankyrins are a family of proteins involved in the targeting of ion channels and transporters to the membrane and are implicated in human heart disease [1,2]. Ankyrin-R (AnkR, encoded by *ANK1*), ankyrin-B (AnkB, encoded by *ANK2*), and ankyrin-G (AnkG, encoded by *ANK3*) are expressed as several isoforms in both excitable and non-excitable cells. Canonical ankyrins have four domains: membrane-binding domain (comprising 24 ANK repeats), a spectrin-binding domain, a death domain, and a C-terminal domain. Canonical AnkR is ~220 kD and was originally identified as an integral part of the erythrocyte cytoskeleton [3,4,5,6], where mutations have been associated with Hereditary Spherocytosis [7,8,9]. A small isoform (sAnkR 1.5, ~20 kD) has been predominantly studied in striated muscle [10,11,12]. sAnkR 1.5 binds to obscurin within the sarcomere and the sarcoplasmic reticulum. While the canonical AnkR isoform has not been studied in striated muscle, besides mouse C2C12 (muscle) cells [13], it was originally identified in erythrocytes and was recently shown to play a role in the anchoring of K_v_3.1b and K_v_3.3 channels in the membrane of neurons [3,14]. While AnkB and AnkG in cardiac disease have been extensively studied, the potential role of cardiac AnkR remains relatively unknown [9,13].

According to a multi-omics post hoc analysis integrating a Genome-Wide Association Study, as well as DNA methylation and gene expression data from the Framingham Heart Study [1], AnkR is significantly associated with diastolic dysfunction, left ventricular remodeling, and heart failure with preserved ejection fraction (HFpEF) [1]. Here, we sought to determine the expression and function of AnkR in the heart. We report that sAnkR is the predominant isoform in cardiomyocytes and canonical AnkR is the predominant isoform in cardiac fibroblasts. Cardiac fibroblasts display perinuclear and cytoplasmic expression of canonical AnkR that has co-localization with the Golgi apparatus. Utilizing conditional AnkR knock-out mice, we sought to investigate the role of AnkR in fibroblast activation and function. Our findings demonstrate that deletion of AnkR in activated fibroblasts alters the composition of collagen fibers and impairs fibroblast function.

## 2. Results

### 2.1. Ankyrin-R Is Expressed in the Heart with Cell-Type Specific Isoform Expression

Using a floxed *Ank1* (*Ank1*^fl/fl^) mouse model, we determined the specificity of our custom anti-AnkR antibody in isolated cardiac fibroblasts. Multiple bands appear on immunoblots near 220 kDa; however, only the lower band disappears after exposure to Adenovirus-Cre, indicating that this band is specific to AnkR (Figure 1A). AnkR is expressed in the total heart lysate as well as the liver, brain, spleen, and kidneys (Figure 1B). We determined the isoform-specific expression between the two major cell types in the murine heart at the mRNA level. Adult cardiomyocytes predominantly express s*Ank1*, while fibroblasts express canonical *Ank1* mRNA (Figure 1C).

To determine if AnkR expression changes upon cardiac stress, we performed a transaortic constriction (TAC) pressure overload model. Male and female C57BL/6 mice underwent either sham or TAC surgery and echocardiographic parameters were measured every two weeks. After six weeks, ejection fraction was significantly reduced in TAC mice versus sham (Appendix A), and cardiac myocytes and fibroblasts were isolated. Immunoblot analysis from both sham and TAC mouse-isolated cells displayed a similar trend to qPCR from wild-type animals, where myocytes express sAnkR (~20 kDa) and fibroblasts express canonical AnkR (~220 kDa) (Figure 1D,E), demonstrating cell-type isoform expression. Interestingly, there was also evidence of a giant AnkR (Figure 1F,G). Increased expression levels of *Rcan1* and *Col1a1* in cardiomyocytes confirm molecular changes associated with TAC (Figure 1H,I). Our novel findings demonstrate that cardiac *Ank1* is expressed as sAnkR in cardiomyocytes and canonical AnkR in cardiac fibroblasts.

After determining the expression pattern of AnkR, we assessed AnkR localization in the murine myocardium. AnkB/G have compartmentalized expression within cardiomyocytes at the intercalated disc (AnkG) and T-tubules (AnkB); however, the expression pattern of ankyrins in fibroblasts, let alone AnkR, has yet to be studied. Immunostaining of AnkR with a ubiquitous membrane marker, WGA-488, demonstrated that AnkR is expressed intracellularly and at the cell membrane (Figure 1J).

### 2.2. Fibroblasts Express Canonical AnkR in the Membrane, Nuclear and Cytoplasmic Compartments

To further assess localization of AnkR, isolated cardiac fibroblasts from wild-type mice were co-immunostained with αSMA, a cardiac fibroblast activation marker, and AnkR (Figure 2A). Diffuse expression of AnkR throughout the cytoplasm (top panel) with a strong perinuclear and nuclear signal (bottom panel) was observed. To confirm these findings and assess the distribution of AnkR, cardiac fibroblasts were fractionated according to subcellular compartment and AnkR was detected by immunoblot. AnkR was observed in most cellular compartments, with the strongest expression coming from the membrane and cytoplasmic fractions (Figure 2B). Since AnkR had strong perinuclear localization and was detected in both the soluble nuclear and cytoplasmic fractions, co-immunostaining was performed with AnkR and trafficking proteins. Immunostaining revealed co-localization between AnkR and TGN38, the trans-Golgi network protein 38, around the nucleus and throughout the cytoplasm (Figure 2C). Multiple studies in the 1990s identified a Golgi-specific ankyrin, earlier termed AnkG119, utilizing an antibody raised against erythrocyte ankyrin, and demonstrated that it could bind to the β1Σ∗ spectrin isoform [13,15,16,17].

### 2.3. Loss of Canonical AnkR Reduces Fibroblast Activity

A contractile assay was used to assess fibroblast activity in the presence or absence of AnkR. A homozygous *Ank1*-floxed mouse model [3] (*Ank1*^fl/fl^) and adenovirus-Cre (Ad-Cre) were used to eliminate AnkR in fibroblasts (Figure 1A). The loss of AnkR results in a significant reduction in contractility after 24 h compared to control cells (Figure 3). While additional studies are required, it is plausible that AnkR loss impacts fibroblast contractility and may play a role in ECM regulation.

### 2.4. Loss of AnkR in Activated Fibroblasts In Vivo Increases Left Ventricular Anterior Wall Thickness

To determine whether the loss of AnkR in activated fibroblasts would alter the fibrotic activity in vivo, *Ank1* was knocked out in activated fibroblasts through the generation of a mouse model using *Ank1*^fl/fl^ mice and a tamoxifen-inducible Cre mouse under the control of the periostin promoter *(Postn*^MCM^; *Ank1*-ifKO) [18] (Figure 4A). To achieve knockout of AnkR in activated fibroblasts, mice were placed on a tamoxifen chow to induce Cre recombinase expression and implanted with osmotic pumps containing 1.5 µg/g angiotensin II and 50 µg/g phenylephrine to activate cardiac fibroblasts (Control PBS *n* = 6, Control AngII/PE *n* = 14; *Ank1*-ifKO PBS *n* = 6, *Ank1*-ifKO *n* = 10) (Figure 4B) [18]. This model has similarities with the development of heart failure with preserved ejection fraction [19,20], where AnkR has been previously associated with HFpEF [1]. After two weeks, electrocardiography (ECG), echocardiography (ECHO), body weight, tibia length, and cardiac tissues were collected (Figure 4B). As expected, AngII/PE treatment significantly decreased left ventricular internal diameter during diastole (LVIDd) and end diastolic volume (EDV) in control mice (PBS *n* = 6, AngII/PE *n* = 14; Figure 5A). Interestingly, a similar change was not observed in *Ank1*-ifKO mice (PBS *n* = 6, AngII/PE *n* = 10; Figure 5A); however, these mice displayed increased thickness of the left ventricular anterior wall during diastole and systole (LVAWd/s) that was not observed in the controls (Figure 5A). Despite significant structural changes present in both control and *Ank1*-ifKO mice treated with AngII/PE, ejection fraction remained statistically unchanged, supporting previous work that suggests that the AngII/PE model recapitulates HFpEF [19,20]. These measurements are supported by representative M-mode echocardiograms showing 2-week PBS control and *Ank1*-ifKO mice still resembling baseline measurements and the difference between the control and *Ank1*-ifKO mice treated with AngII/PE for 2 weeks (Figure 5B). It has been demonstrated that after 10 days of chronic AngII/PE infusion, mature collagen fibers (type I) are present and initial inflammation due to the presence of neutrophils returns to baseline [20]. Fourteen days of chronic infusion was sufficient to induce similar structural characteristics attributed to a HFpEF model and allowed for the investigation of the loss of AnkR during the fibrotic remodeling phase instead of the initial inflammatory phase [20]. Both male and female mice were used throughout the experiment, and post hoc analysis showed no significant difference in cardiac parameters between sexes.

### 2.5. Ank1-ifKO Mice Exhibit Fibrosis Deposition Changes

To assess the in vivo effects of AnkR loss in activated fibroblasts, hearts from *Ank1*-ifKO mice treated with PBS or AngII/PE were assessed for collagen deposition (Figure 5C,D). Staining with Masson’s Trichrome did not reveal any differences in fibrosis (Figure 5C). Despite no alterations in total collagen deposition, collagen type was assessed with picrosirius red and polarized light microscopy to visualize the quality of collagen fibers since collagen type III appears greenish-yellow and collagen type I appears reddish-orange under these conditions. A significant decrease was observed in the amount of reddish-orange collagen type I in the *Ank1*-ifKO mice treated with AngII/PE compared to control AngII/PE mice as well as *Ank1*-ifKO mice treated with PBS (Figure 6), suggesting that the loss of AnkR blunts the maturation of collagen fibers within the myocardium. This process is an important response to stress as the heart transitions from the reactive/inflammatory phase containing predominantly collagen type III (green) fibers through the proliferative and remodeling phases containing predominantly collagen type I (red) fibers.

To determine if the loss of AnkR in activated fibroblasts results in electrical abnormalities after AngII/PE challenge, subsurface electrocardiograms were performed. Control mice treated with AngII/PE had a statistically significant increase in QRS interval and ST height elevation, while no such change was observed in *Ank1*-ifKO mice (Appendix A). These results demonstrate that control mice have delayed depolarization and their hearts exhibited ischemic-related ST height alterations, while hearts from mice lacking AnkR in activated fibroblasts were not significantly impacted from the AngII/PE challenge.

## 3. Discussion

Here, we identify the expression of canonical AnkR in cardiac fibroblasts. At the cellular level, AnkR is expressed in the cytoplasmic, nuclear, and perinuclear space of these fibroblasts. Cardiac fibroblasts lacking AnkR had reduced fibroblast activity. At the whole animal level, mice lacking AnkR in activated fibroblasts displayed increased cardiac thickness in the LVAW, with increased type I collagen fibers under normal conditions, but were significantly reduced after AngII/PE treatment. Additionally, the loss of AnkR in fibroblasts prevented heart failure-induced electrical abnormalities seen in the controls, supporting a new role for AnkR in the heart for the potential regulation of ECM deposition in heart failure.

AnkR colocalized with the cellular membrane in cardiac tissue. A previous study demonstrated a distinct banding pattern for sAnkR in cardiomyocytes that was not observed here [21], which may be a limitation of our antibody or indicate non-specific staining with commercial antibodies. In cardiac fibroblasts, AnkR is diffusely expressed and has a strong perinuclear and nuclear signal. The perinuclear space serves several functions, including traffic regulation between the nucleus and cytosol, nuclear structure and support, and modulation of signaling pathways between the nucleus and other organelles within the space, such as the endoplasmic reticulum, mitochondria, and trafficking vesicles [22,23]. Ankyrins bind to proteins present within the perinuclear space, including the EH domain, containing proteins (EHD1-4) [24]. Antisera raised against erythrocyte ankyrin has been shown to stain the Golgi apparatus [16], suggesting that AnkR in isolated fibroblasts may bind Golgi machinery. A model for this interaction between ankyrin, spectrin, and the Golgi has been previously proposed [25,26]. Here, cardiac fibroblasts demonstrate co-localization between AnkR and trans-Golgi network protein 38 (TGN38) (Figure 2).

Dysfunctional ECM deposition is one hallmark of HFpEF and is predominantly orchestrated by fibroblasts [27]. Deletion of *Ank1* in cardiac fibroblasts in vitro demonstrated reduced collagen compaction in comparison to the control (Figure 3). This suggests that in the absence of AnkR, cardiac fibroblasts have decreased activity or less ECM deposition to stiffen/compact the collagen matrix. Since we observed co-localization of AnkR with Golgi machinery, AnkR may have a role in the secretion of matrix proteins via vesicle trafficking. Ankyrins biologically serve as molecular adapters of membrane-bound proteins to the cytoskeleton and as platforms to which other adapter proteins may bind [2]. Since the loss of AnkR in fibroblasts resulted in alterations to collagen maturation and thickening of the left ventricular anterior wall during diastole and systole, we hypothesize that fibroblasts have a blunted activation response. Additional studies are warranted, but AnkR may potentially regulate the surface expression of proteins that facilitate fibroblast activation and are associated with extracellular matrix deposition. Conversely, AnkR has been proposed to associate with Golgi machinery. It is possible that the loss of AnkR leads to a blockage of collagen-loaded Golgi vesicles.

Corresponding with the multi-omic analysis from the Framingham Heart Study [1], we observed significant increases in LVAWs, LVAWd, and LVPWd in *Ank1*-ifKO mice compared to control animals also treated with AngII/PE. Further, with AngII/PE treatment, fibrosis deposition trended towards a significant increase within both perivascular and interstitial compartments of *Ank1*-ifKO over control hearts. Despite structural remodeling, EF was unchanged across all groups. As expected, AngII/PE treatment significantly decreased LVIDd and EDV in the control mice [20]. Interestingly, a similar change was not observed in *Ank1*-ifKO mice; however these mice did display increased thickness of the left ventricular anterior wall during diastole and systole (LVAWd/LVAWs), which was not observed in the control mice (Figure 5). Cardiac fibrosis can be relatively split into three phases: a reactive/inflammatory phase, the proliferation phase, and the maturation phase. In the presented paradigm, these AngII/PE-treated mice are likely captured during the in vivo cellular processes of the first two phases. It is possible that the maturation phase of cardiac fibrosis is not captured in our in vivo analysis and whether the presence of AnkR is necessary for this transition is not understood. Further, instead of being important in the pathology of HFpEF, it would be interesting to investigate whether AnkR is vital for the development and maturation of fibroblasts during development. Future studies are warranted using the *Tcf21*-Cre to elucidate the contribution of AnkR in activated and non-activated cardiac fibroblasts. Finally, we have demonstrated that deletion of AnkR ameliorates the delayed depolarization and ST height elevation observed after AngII/PE treatment.

Here, we present a foundation for future investigations into cardiac AnkR. While the contribution of fibroblasts in the pathophysiology of diastolic dysfunction and HFpEF is an active area of research, the possible role of ankyrins in fibroblasts has yet to be elucidated. The focus of this work was on cardiomyocytes and cardiac fibroblasts, and the contribution of AnkR in other cardiac cell types, including specialized cardiomyocytes, macrophages, and endothelial cells, have not yet been explored. Mouse models of HFpEF are controversial, as many groups have recognized that HFpEF is a multi-organ dysfunction syndrome with metabolic and cardiac deficits [28]. As such, another model is aged mice on a high-fat diet treated with a nitric oxide inhibitor [20], and it would be valuable to repeat these studies in this model that reflects more similarities to human HFpEF presentation. Additional studies are warranted to elucidate how AnkR regulates fibrotic activity and interstitial remodeling to support the development of novel therapeutic interventions.

## 4. Materials and Methods

### 4.1. Animal Studies

All animal experiments were performed according to procedures approved by the Ohio State University’s Institutional Animal Care and Use Committee (# 201100000034). Adult (3–4 months) C57BL/6J male and female wild-type (WT) mice, *Ank1*^fl/fl^ mice, and *Postn*^MCM^ mice were used for experiments [3]. *Postn*^MCM^ mice were cross-bred with *Ank1*^fl/fl^ mice to obtain tamoxifen-inducible *Ank1*-ifKO mice. Adult male and female *Ank1*-ifKO or control (*Ank1*^fl/fl^, Cre-) mice were fed a diet containing 400 mg/kg tamoxifen citrate (Envigo/Inotiv, Indianapolis IN, USA TD.130859) during the entirety of the two-week osmotic pump experiments.

### 4.2. Osmotic Pump Surgeries

All mice used for surgery were aged to 12–14 weeks, weighing at least 20 g. Surgeries were conducted as previously described [18]. Briefly, under 2.0% isoflurane anesthesia, micro-osmotic pumps (Azlet Model 1002) were inserted subcutaneously, delivering 1.5 µg/g/day angiotensin II (Sigma-Aldrich, Burlington, MA, USA, A9525) and 50 µg/g/day phenylephrine hydrochloride (Sigma, Burlington, MA, USA, P6126) for two weeks. Control animals were treated with saline-loaded osmotic pumps. Mice were euthanized by CO_2_ asphyxiation and bilateral pneumothorax. Due to tamoxifen chow administration post-surgery, some animals were lost after surgery, leading to unequal group numbers (lost: AngII/PE-treated controls: *n* = 6, *Ank1*-ifKO *n* = 3. Multiple studies were combined for this data with the goal of *n* = 12 for *Ank1^f^*^l/fl^ controls and ifKO, and *n* = 6 per PBS treated. Animals were divided for downstream experiments: immunoblot, sectioning, cell isolations.

### 4.3. Primary Cell Isolations

Mouse cardiac fibroblasts (CFs) and cardiomyocytes were isolated from left and right ventricles under sterile conditions as described [29]. Briefly, for CF harvests, mouse tissue was minced into 8–12 pieces and enzymatically digested at 4 °C overnight in HBSS (Thermo Fisher, Waltham, MA, USA 24020-117) containing 2 mg/mL of trypsin (1:250 powder Gibco/Thermo Fisher, Waltham, MA, USA, 27250-018) with gentle rocking. The next day, cardiac tissue was serially digested with HBSS containing 1 mg/mL collagenase type II (Worthington Biochemical, Freehold, NJ, USA LS004177) at 37 °C with orbital shaker agitation set to 125 rpm. Cell suspension was collected in culture media after each digestion and ultimately centrifuged with the pellet resuspended in culture media: DMEM, high glucose/pyruvate (Thermo Fisher, 11995073) supplemented with 10% heat-inactivated fetal bovine serum (Thermo Fisher, 16140071), and 1% penicillin-streptomycin (10,000 U/mL) (Thermo Fisher, 15140122). Cardiac fibroblasts were cultured up to P2. Adult ventricular cardiomyocytes were prepared as previously described [30,31].

### 4.4. Collagen Gel Formation and Contraction Assay

Type I rat-collagen gels (2 mg/mL) were prepared by mixing 10× PBS, sterile H_2_O, acidic rat tail collagen, and 1 M NaOH as previously described [29]. Cells were added (150,000 cells/mL) and mixed before gelation. Cell–collagen mixtures were cast into 24-well culture plates and incubated at 37 °C in 5% CO_2_ for 1 h. After incubation, 1 mL of medium was added, and the gels were released from the wells. For the adenovirus experiments, using Ad-Cre or Ad-GFP (Vector Biolabs, Malvern, PA, USA, #1060 and #1700), the virus was added at an MOI of 100 in serum free media directly to the collagen solution containing cells. Images were obtained prior to the start of the experiment and at 24 h intervals thereafter for 72 h. Images were analyzed using ImageJ software (https://imagej.net/nih-image/, accessed on 23 July 2024). As previously described, isotropic compaction was assumed to measure the volume ratio of gels before and after compaction [32]. Gels without cells served as controls. Experiments were conducted in technical triplicates.

### 4.5. Echocardiography

Transthoracic echocardiography was performed using a GE Logiq e system with a L10–22 (mHz) transducer. Mice were anesthetized with 2.0% isoflurane in 95% O_2_ and 5% CO2 at ~0.8 L/min while maintenance of sedation was continued with ~1% isoflurane throughout the recording process. Following hair removal, recordings of the left ventricle (LV) were collected in a long axis view, with contractile parameters and chamber dimensions collected using the M-mode function in the short axis view. Parameters analyzed in short axis M-mode include interventricular septum left ventricular anterior wall thickness during diastole and systole (LVAWd/s), left ventricular internal diameter at diastole and systole (LVIDd/s), end diastolic volume (EDV), left ventricular posterior wall end diastole and systole (LVPWd/s), end systolic volume (ESV), ejection fraction (EF), fractional shortening (FS), and heart rate (HR).

### 4.6. Electrocardiogram

Mice were anesthetized initially with 2% isoflurane in 95% O_2_ and 5% CO_2_ at ~0.8 L/min prior to subsurface ECG lead placement, while maintenance of sedation was continued at ~1% isoflurane throughout the recording process. Baseline and 2-week recordings were completed for 10 min. Analysis was completed with LabChart 8 (AD Instruments, Sydney, Australia).

### 4.7. Immunoblotting

Protein lysates were quantified using Pierce BCA Protein Assay Kit (Thermo Fisher, 23227). Tissue collection was preceded by a thorough 50 mL PBS syringe perfusion through the left ventricle of the heart and subsequent submersion in ice-cold PBS to remove residual blood. Tissue homogenization was performed using bead homogenization (Precellys 24, Bertin Instruments, Rockville, MD, USA) and cellular lysate was extracted using a 26G needle with repeated aspiration to apply shear stress. An amount of 35–50 µg of protein lysate mixed with 1× Laemmli buffer supplemented with 1% BME and 1× Halt Protease and Phosphatase inhibitor cocktail (Thermo Fisher, 78442) were loaded into 4–15% precast Mini-PROTEAN TGX Stain-Free gels (Bio-Rad, Hercules, California USA 4568084). Gels were then transferred to 0.2 µm Nitrocellulose membranes utilizing a Trans-Blot Turbo Transfer System (Bio-Rad, 17001918) at mixed molecular weight setting. Images were taken for unstained loading controls (labelled total protein). Membranes were subsequently blocked in 1% Blocker Casein in PBS (Thermo Fisher, 37528) for 1 h at room temperature. Following blocking, membranes were incubated with primary antibody (custom AnkR antibody 1:1000) overnight at 4 °C and then incubated with either donkey anti-rabbit (1:2500, Jackson ImmunoResearch Laboratories, West Grove, PA, USA 711-035-152) or donkey anti-mouse (1:2500, Jackson ImmunoResearch Laboratoreis, 715-035-150) HRP secondary antibodies for 2 h at room temperature. Chemiluminescence of HRP was developed using SuperSignal West Femto Maximum (Thermo Fisher, 34096, 1:1). Densitometry was performed using ImageJ software (NIH, Bethesda, MD, USA). Custom AnkR antibody was generated by Covance Antibody Services utilizing the AnkR-specific C-terminal peptide: ATEHDTMLEMSDMQ in rabbits. Validation of the antibody was tested using Ad-Cre knockout in cardiac fibroblasts (Figure 1A and Figure 4).

### 4.8. Immunofluorescence

Whole hearts were isolated from adult WT mice and immediately rinsed in cold PBS before embedding into optimal cutting temperature (OCT) compound (Fisher Scientific/Thermo Fisher, USA, 23-730-571) and frozen in liquid nitrogen vapor. Hearts were cryosectioned at 10 µm thickness. Isolated cardiac fibroblasts were cultured on glass coverslips and fixed with 1:1 acetone:methanol for 10 min and incubated in blocking solution (PBS with 3% fish gelatin, 0.75% Triton-X 100 (10%), 1% DMSO) for 1 h. Cells were then incubated in primary antibody (custom AnkR antibody (1:300); WGA-488 5 µg/mL Thermo Fisher, W11261; TGN38 1:1000, Novus Biologicals, Centennial, CO USA NB300-575; αSMA, 1:500 Sigma-Aldrich A2547-100UL) at 4 °C overnight. The next day, cells were washed with blocking solution and incubated for 2 h at room temperature with secondary antibody. Concurrently, no primary and no secondary controls were used. Sections and isolated fibroblasts were covered with ProLong Glass Antifade Mountant with NucBlue Stain (Thermo Fisher, P36981) and allowed to cure overnight before imaging with a confocal microscope (LSM 510 Meta, Zeiss, Oberkochen, Germany).

### 4.9. Transcript Analysis

Total RNA was isolated using Trizol (Thermo Fisher, 15596026). Tissues and cells were bead-homogenized at 4 °C in Trizol. RNA concentrations and purity were assessed using a Nanodrop 1000 (Thermo Fisher, Wilmington, DE, USA). mRNA up to 2 µg was reversed transcribed using SuperScript IV VILO Master Mix (Thermo Fisher, 11756050). qPCR was performed on cDNA utilizing PowerUp SYBR Green Master Mix (Thermo Fisher, A25742) Digital droplet PCR was performed using a BioRad QX200 Digitial Droplet PCR system using EvaGreen. HPRT was used as the internal control and a total of 100 ng of cDNA was used to detect low abundance *Ank1*.

### 4.10. Tissue Histology and Staining

Whole hearts were isolated from adult WT, control, or *Ank1*-ifKO mice at baseline and 2-week time points. Hearts were briefly rinsed in cold PBS, and either placed directly in OCT (staining) or fixed for 3.5 h in 4% PFA at 4 °C, rinsed with PBS, and cryoprotected in 30% sucrose overnight before embedding in OCT (Trichrome experiments). Heart sections were collected using a cryostat at 10 µm for either four-chamber or two-chamber views. For staining, sections were allowed to come to room temperature for 30 min before being incubated in ice-cold acetone for 5 min, air-dried, and coated in blocking solution (PBS with 3% fish gelatin, 0.75% Triton-X 100 (10%), and 1% DMSO) for 1 h. For trichrome, sections were allowed to come to room temperature for 30 min, then pre-fixed in 10% formalin at room temperature for 1 h. Sections were then placed into Bouin’s solution (Sigma-Aldrich HT10132) for 15 min at 62 °C, rinsed in running tap water and placed into Weigert’s Hematoxylin (Sigma-Aldrich, HT107) stain for 5 min followed by an additional rinse. The sections were then placed into Biebrich Scarlet-Acid Fuchsin (Sigma-Aldrich, HT151) for 15 min, rinsed, incubated in Phosphomolybdic/Phosphotungstic Acid (Sigma-Aldrich HT153) for 10 min, and directly placed into Aniline Blue (Sigma-Aldrich B8563) stain for 3 min. Finally, sections were rinsed and immersed in 1% acetic acid for 3 min, and immediately dehydrated in 100% ethanol for 1 min/3 changes and cleared in xylene for 1 min/3 changes and allowed to air dry before permanent mounting and coverslip placement. Images were obtained using a confocal microscope for staining (LSM 510 Meta, Zeiss, Oberkochen, Germany) at identical protocol settings, with the observer blinded to the genotype. Images were obtained using an EVOS M7000 (Thermo Fisher, USA) for trichrome. Fibrosis analysis of trichrome images was quantified using the method as previously described [33].

### 4.11. Statistics

Data are presented as mean ± S.E.M. For the comparison of multiple groups, we performed a two-way ANOVA with Tukey’s multiple comparison test or three-way ANOVA. When comparing two groups, unpaired *t*-tests were used. GraphPad Prism 10 was used for statistical analysis and graph creation.

## Figures and Tables

**Figure 1 ijms-25-08403-f001:**
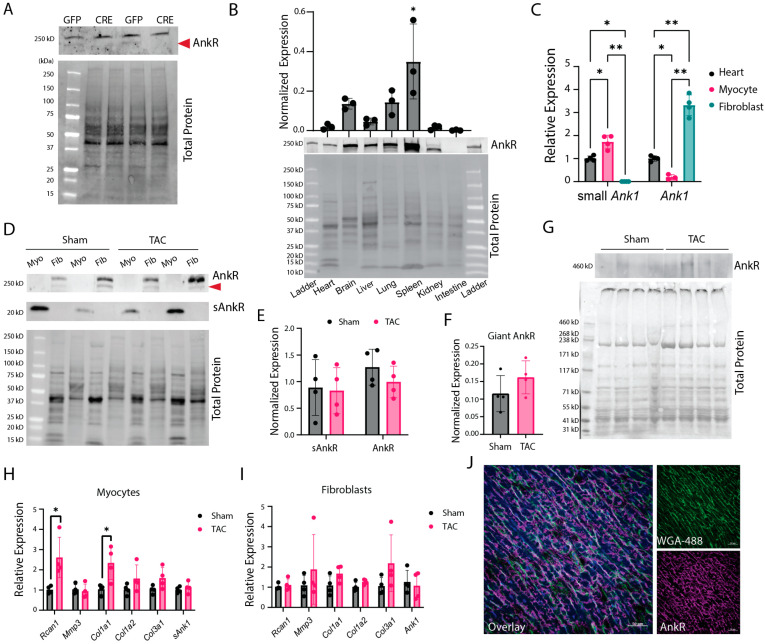
Ankyrin-R is expressed in the heart and differentially regulated based on cell type. (**A**) Isolated CFs from *Ank1*^fl/fl^ mice treated with Adenovirus-Cre to knockdown AnkR (or Ad-GFP control). With Cre excision, the bottom band is no longer present, showing successful knockdown, *n* = 4. (**B**) Wild-type mouse tissue immunoblot image assessing AnkR expression across heavily perfused tissues, with high expression in the spleen and identifying AnkR in the heart, *n* = 3. One-way ANOVA showed that the spleen had significantly more AnkR * *p* < 0.05. (**C**) *Ank1* expression in isolated myocytes and fibroblasts, *n* = 4, two-way ANOVA, Tukey’s multiple comparisons test * *p* < 0.05, ** *p* < 0.0001. (**D**,**E**) Isolated cardiomyocytes or fibroblasts from wild-type mice with sham or TAC surgery show sAnkR is predominantly expressed in cardiomyocytes, while canonical protein is expressed in cardiac fibroblasts, *n* = 4, unpaired *t*-test. (**F**,**G**) Immunoblot and quantification from wild-type mice with sham or TAC surgery showing a giant AnkR isoform, *n* = 4 unpaired *t*-test. (**H**,**I**) qPCR results showing increased *Rcan1* and *Col1a1* in isolated myocytes (**H**) from TAC mice, but no changes in fibroblasts (**I**). Unpaired *t*-tests per gene set, *n* = 4 * *p* < 0.05. (**J**) AnkR co-localizes with WGA-488 at the membrane, *n* = 3. Scale bar 50 µm.

**Figure 2 ijms-25-08403-f002:**
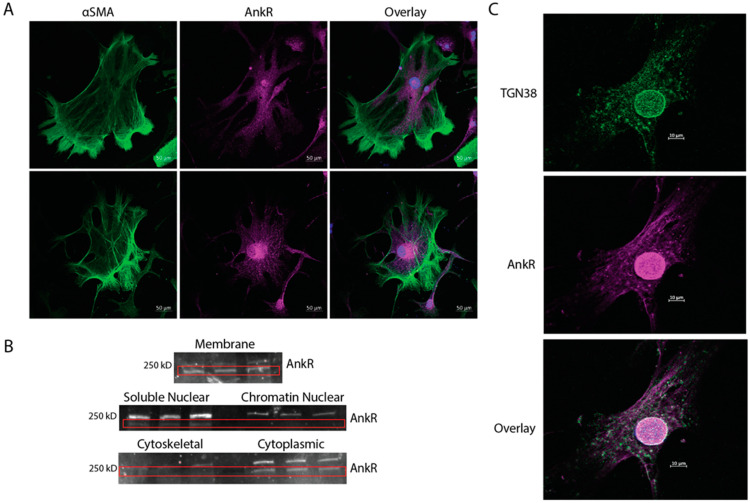
Fibroblasts express AnkR at the membrane and cytoplasm. (**A**) Isolated cardiac fibroblasts (CFs) from wild-type mice were stained with αSMA and AnkR, identifying strong cytoplasmic and perinuclear staining, *n* = 3. (**B**) Subcellular fractionation of isolated CFs displayed canonical AnkR in most cellular compartments, but predominantly from the membrane and cytosolic fractions, *n* = 3, scale bar = 50 µm. (**C**) Isolated CFs co-immunostained with TGN38 and AnkR showing colocalization in white, *n* = 3, scale bar = 10 µm.

**Figure 3 ijms-25-08403-f003:**
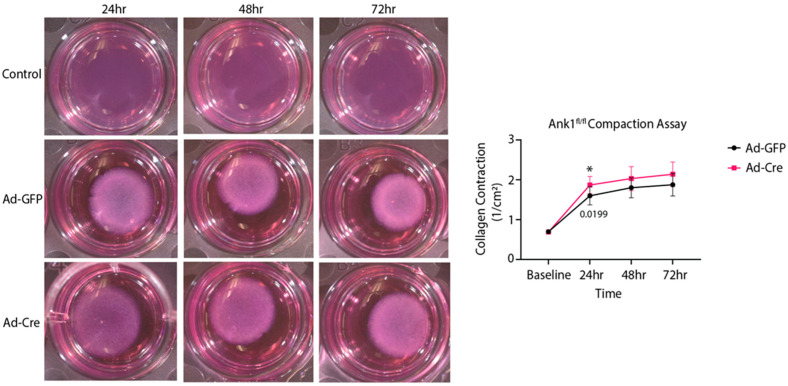
Isolated cardiac fibroblasts from *Ank1*^fl/fl^ mice exhibit less compaction. Isolated *Ank1*^fl/fl^ cardiac fibroblasts were plated and treated with Ad-Cre or Ad-GFP. The collagen disks (2 mg/mL) were measured every 24 h and quantified in the graph on the right; *N* = 3, *n* = 3, unpaired *t*-tests at each time point to assess genotype effect, * *p* < 0.05, mean ± SEM.

**Figure 4 ijms-25-08403-f004:**
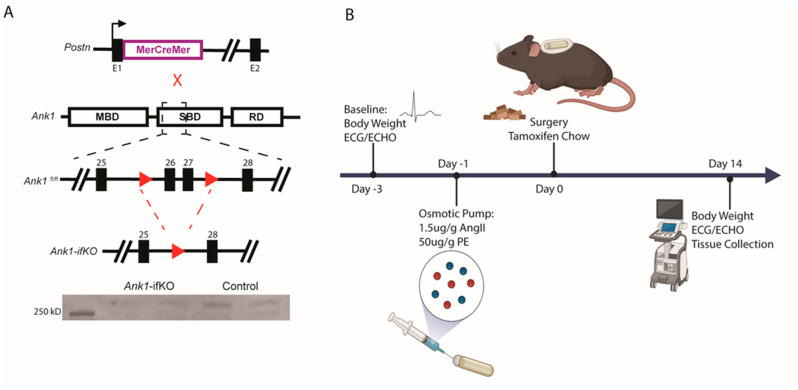
Mouse generation and experimental design. (**A**) The *Postn*^MerCreMer^ mice were crossed with *Ank1*^fl/fl^ mice to produce an inducible *Ank1*-ifKO model in cardiac fibroblasts. Exons 26 and 27 are excised in Cre recombinase conditions for AnkR deletion, *n* = 4. (**B**) Baseline body weight, ECG, and ECHO were recorded prior to the study start, then the osmotic pumps were loaded and allowed to equilibrate in solution the day before implantation. Surgery was completed to insert the osmotic pumps, and tamoxifen chow was given to the mice and continued throughout the study to induce Cre activation. After 2 weeks, bodyweight, ECG, ECHO, and tissues were collected.

**Figure 5 ijms-25-08403-f005:**
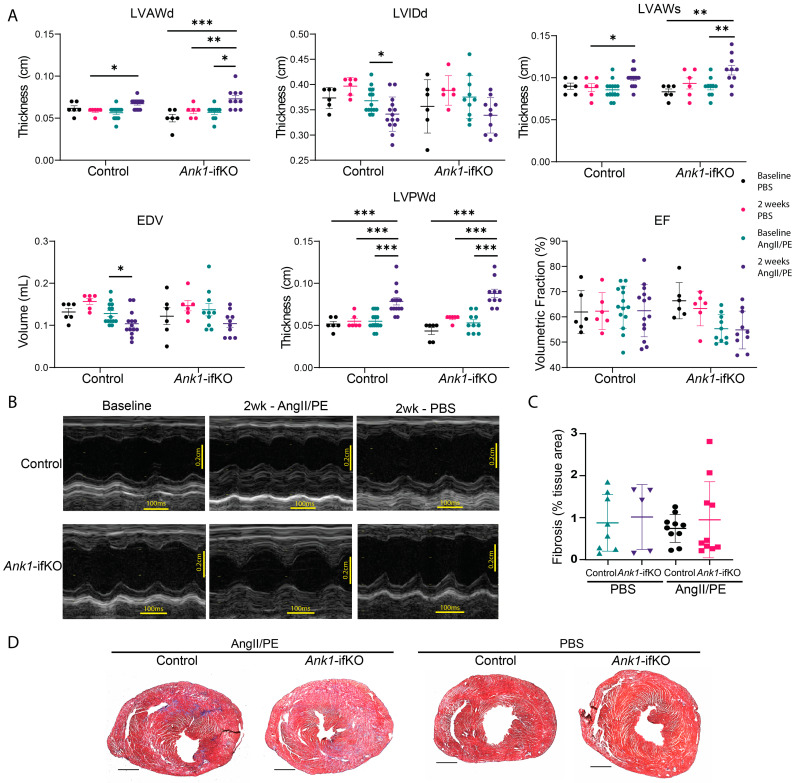
Echocardiographic results display structural changes in *Ank1*-ifKO mice. (**A**) ECHO results after two weeks of AngII/PE or PBS treatment in *Ank1*^fl/fl^ controls and *Ank1*-ifKO: LVAWd (left ventricular anterior wall during diastole), LVIDd (left ventricular diameter during diastole), LVAWs (left ventricular anterior wall during systole), EDV (end-diastolic volume), LVPWd (left ventricular posterior wall thickness during diastole), EF (ejection fraction). Mice had preserved ejection fraction despite structural changes. Control PBS *n* = 6, Control AngII/PE *n* = 14; *Ank1*-ifKO PBS *n* = 6, *Ank1*-ifKO *n* = 10, three-way ANOVA, mean ± SEM * *p* < 0.05, ** *p* < 0.001, *** *p* < 0.0001. (**B**) Representative ECHO imaging displaying M-mode to measure these parameters from a two-chamber view. Scale bars: 100 ms and 0.2 cm. (**C**) Quantification of trichrome staining in (**D**) showing no difference in fibrosis staining after 2 weeks. three-way ANOVA, mean ± SEM. *Ank1*^fl/fl^ control, PBS *n* = 8, AngII/PE *n* = 10; *Ank1*-ifKO, PBS *n* = 5, AngII/PE *n* = 10. (**D**) Representative trichrome images of the heart (two-chamber view) between groups.

**Figure 6 ijms-25-08403-f006:**
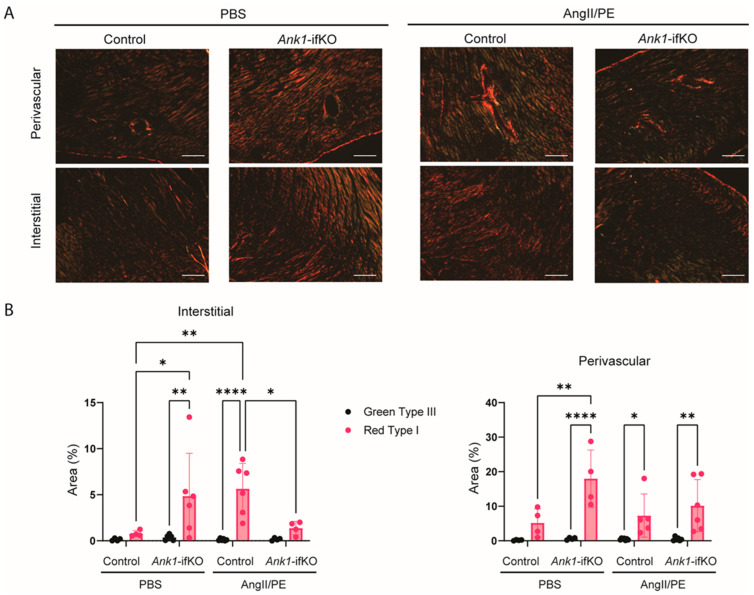
*Ank1*-ifKO mice exhibit fibrosis deposition changes. (**A**) Representative images of Picrosirius Red staining of the myocardium under polarized light in the different groups showing collagen type I fibers (red) and collagen type III fibers (green). (**B**) Quantification of Red type I and Green type III collagen fibers in the interstitial and perivascular space. *Ank1*-ifKO had significantly less Red Type I fibers after AngII/PE treatment in the interstitial space. Control PBS *n* = 4, *Ank1*-ifKO PBS *n* = 4–6, Control AngII/PE *n* = 6, *Ank1*-ifKO AngII/PE *n* = 4–6. Two-way ANOVA, Tukey’s multiple comparisons test, * *p* < 0.05, ** *p* < 0.001, **** *p* < 0.00001 mean ± SEM.

## Data Availability

The original contributions presented in this study are included in the article/Appendix A. Further inquiries can be directed to the corresponding author. All data have been included in this manuscript.

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
