# Peer review of "Novel Identification of Ankyrin-R in Cardiac Fibroblasts and a Potential Role in Heart Failure"

_ijms, 2024, doi:10.3390/ijms25158403_

Round 1

Reviewer 1 Report

Comments and Suggestions for Authors

The authors present an interesting study examining the role of ankyrin-R in cardiac tissue. The authors show for the first time AnkR in cardiac myocytes and cardiac fibroblasts, and utilise an in vivo model to determine the functional role in cardiac function. In those animals lacking AnkR there was a reduced level of collagen compaction, with those anumals also demonstrating an increase in fibrosis and cardiac readouts. Taken together, this article reveals the significance of this protein in the heart for the first time, earmarking it as a potential therpaeuti target in cardiac related health contexts.

1.      In reviewing the manuscript I made a couple of observations. The authors should consider the following when drafting a suitable resubmission.

2.      For figure 5 data, why does each group have unequal numbers? The authors should justify why there is significant differences between each group in terms of the number of replicates in each.

3.      It would be useful if every figure legend had the n-number included i.e. Figure 6

4.      In what instances was 1% or 10% FBS used in the experiments?

5.      What concentration of rat tail collagen was used for the collagen gel assays?

6.      More details on the custom AnkR antibody should be provided. What is the source of this antibody? How was it validated?

7.      There appears to be many non-specific bands in those westerns submitted with the article.  Did the authors perform any quality controls to ensure those bands they are identifying as their target are indeed those?

Reviewer 2 Report

Comments and Suggestions for Authors

Reviewer report

Novel identification of Ankyrin-R in cardiac fibroblasts and the potential role during heart failure

This is a well-organized study showing the importance of Ankyrin-R heart diseases. The article focuses lights on the how Ankyrin-R is involved in the pathophysiology of the heart failure. Study shows well explained introduction, background and appropriate results. There are few sections where manuscript can be improved.

1. Please represent the entire study in the form of graphical abstract.

2. In normal physiology where does Ankyrin-R is expressed in the body what are the functions and in which pathophysiological conditions levels are altered please elaborate briefly.

3. How this animal model can be correlated to humans in the aspect of findings please explain.

4. What was the effect on the left ventricular anterior wall thickness for systole and diastole, cardiac output, and other systolic parameter please explain.

5. Please specify Whether the diastolic functional parameter of mitral valve E/A ratio was recorded for the study.

Reviewer 3 Report

Comments and Suggestions for Authors

Reviewing the review manuscript entitled, “Novel identification of Ankyrin-R in cardiac fibroblasts and the potential role during heart failure” by Argall AD et al., this is an article focusing on ankyrin-R in cardiac fibroblasts. It is extremely interesting to focus on ankyrin as a mechanism for the development of HFpEF in the cardiac interstitium. The authors need to respond to the following concerns.

In the introduction section, the authors should provide detailed descriptions of the molecular biology of ankyrin, associating with figures.

In your study, knockout of ankyrin-R resulted in impaired interstitial development. The authors should provide the molecular biological mechanisms in the discussion section.

In TAC experiments, ankyrin-R showed no changes in expression. Nevertheless, gene deletion inhibited stromal development and promoted fibrotic activity in fibrotic activity experiments. The authors should describe ankyrin-R activation mechanism.

Is it possible that ankyrin-R inhibition can prevent the development of HFpEF? Should the experimental results be interpreted in this way? This is of great clinical importance.

Round 2

Reviewer 1 Report

Comments and Suggestions for Authors

The authors have suitably addressed my comments.

Reviewer 3 Report

Comments and Suggestions for Authors

This is an acceptable quality. Congrats.